# The Role of High-Resolution Magnetic Resonance Imaging in Cerebrovascular Disease: A Narrative Review

**DOI:** 10.3390/brainsci13040677

**Published:** 2023-04-18

**Authors:** Xiaohui Li, Chengfang Liu, Lin Zhu, Meng Wang, Yukai Liu, Shuo Li, Qiwen Deng, Junshan Zhou

**Affiliations:** 1Department of Neurology, Nanjing First Hospital, Nanjing Medical University, Nanjing 210006, China; 2Department of Neurology, Beijing Tiantan Hospital, Capital Medical University, Beijing 100070, China

**Keywords:** high-resolution magnetic resonance imaging, vessel wall imaging, vulnerable plaques, cerebral arterial disease, technique

## Abstract

High-resolution magnetic resonance imaging (HRMRI) is the most important and popular vessel wall imaging technique for the direct assessment of vessel wall and cerebral arterial disease. It can identify the cause of stroke in high-risk plaques and differentiate the diagnosis of head and carotid artery dissection, including inflammation, Moya Moya disease, cerebral aneurysm, vasospasm after subarachnoid hemorrhage, reversible cerebral vasoconstriction syndrome, blunt cerebrovascular injury, cerebral arteriovenous malformations, and other stenosis or occlusion conditions. Through noninvasive visualization of the vessel wall in vitro, quantified assessment of luminal stenosis and pathological features of the vessel wall can provide clinicians with further disease information. In this report, technical considerations of HRMRI are discussed, and current clinical applications of HRMRI are reviewed.

## 1. Introduction

Cerebrovascular disease is a leading cause of morbidity and mortality worldwide, with stroke being the second most common cause of death globally [1]. A major cause of ischemic stroke is the rupture of atherosclerotic plaques [2]. The degree of luminal stenosis in the head and carotid arteries has been considered the traditional measure of the severity of atherosclerotic disease. However, recent evidence has shown that this criterion alone may not accurately assess the risk of adverse events associated with vulnerable plaques [3]. Therefore, component analysis of atherosclerosis has been utilized to evaluate the extent and mechanism of stenosis and predict the recurrence of ischemic events such as stroke and transient ischemic attack.

High-resolution magnetic resonance imaging (HRMRI) is a non-invasive diagnostic tool that has emerged as a promising technique for evaluating cerebrovascular disease. It can identify stroke mechanisms, determine the extent and pathology of stenosis, and recognize plaque characteristics that cannot be visualized by conventional imaging methods [4,5,6]. HRMRI is performed on high-field strength MRI scanners (usually 3T or higher) with specialized coils and pulse sequences. The main sequence used for HRMRI is the 3D time-of-flight (TOF) magnetic resonance angiography (MRA) sequence, which provides high-resolution images of the cerebral vasculature, including the intracranial arteries and veins. This sequence uses the flow-related enhancement of blood to generate images of the vasculature without the need for contrast agents. In addition to TOF-MRA, other HRMRI sequences, including T1-weighted imaging (T1WI), T2-weighted imaging (T2WI), contrast enhanced T1-weighted imaging (CE-T1WI), and proton density-weighted imaging (PDWI), can provide highly sensitive visualization and quantitative analysis of major tissue components, which are important predictors of plaque vulnerability (Table 1) [7,8,9]. However, not all intracranial atherosclerotic plaques contain these layers or exhibit enhancement [10]. This technology enables the stratification of patients based on their risk profiles, facilitating the selection of appropriate treatment strategies and the evaluation of treatment efficacy in reducing plaque progression. Atherosclerotic plaques are composed of various components such as calcification, necrotic lipid cores, hemorrhagic areas, and fibers. Vulnerable plaques contain a large lipid-rich necrotic core, intraplaque hemorrhage (IPH), or thin or ruptured fibrous caps (FC) that are prone to rupture (Figure 1). High lumen stenosis or occlusion may lead to remote cerebral hypoperfusion, particularly in patients with insufficient collateral blood flow [11,12], contributing significantly to morbidity and mortality [7,13,14]. Progressive thinning of the plaque cap can lead to plaque rupture, acute thrombosis, and luminal obstruction caused by the release of thrombotic material from the plaque. Intraplaque hemorrhage is caused by the rupture of plaque microvessels, resulting in erythrocyte membrane accumulation, cholesterol deposition, macrophage infiltration, enlargement of the necrotic core, and atherosclerotic growth and plaque instability [15,16,17,18]. The abundance of lipids in the necrotic core of the plaque is also an indicator of vulnerable plaques [19,20,21,22]. The fibrous cap is the connective tissue layer covering the necrotic core of lipids. The rupture of a thin fibrous cap can expose the thrombosis lipid core to the blood circulation, leading to thromboembolism. However, the thick fibrous cap is not easy to break [19,23,24,25,26]. HRMRI has been used to determine the contribution of ipsilateral carotid plaques to neurological ischemic events at varying degrees of stenosis, and it has also proved useful in predicting the risk of vascular recurrence or neurological events in asymptomatic plaques [27]. Furthermore, HRMRI can aid in the analysis of other conditions involving stenosis or occlusion, such as head and carotid artery dissection, inflammation, Moya Moya disease (MMD), cerebral aneurysm, vasospasm after subarachnoid hemorrhage (SAH) [28], reversible cerebral vasoconstriction syndrome [29], blunt cerebrovascular injury (BCVI) [30], cerebral arteriovenous malformations, and other stenosis or occlusion conditions [31,32]. Overall, the application of HRMRI has the potential to significantly enhance stroke prevention research. We searched the terms “high-resolution magnetic resonance imaging”; “magnetic resonance imaging (MRI)”; “carotid plaques”; “cerebral arteries”; “intracranial atherosclerotic stenosis”; “vulnerable plaques”; “vessel wall”; “stroke”; “clinical relevance”; “histology” on PubMed for various articles about the research and development of HRMRI and imaging methods for cerebral atherosclerotic stenosis. Collect extracts from all articles by title and abstract for discussion and analysis. No date, patient population, or article type was excluded.

## 2. HRMRI of Extracranial Carotid Atherosclerosis

HRMRI has become an increasingly valuable tool in the diagnosis and management of extracranial carotid atherosclerosis. Extracranial carotid atherosclerosis refers to the buildup of plaque in the carotid arteries, which can lead to stenosis, or narrowing, of the vessels and increase the risk of stroke. For patients with extracranial carotid stenosis, the HRMRI definition of vulnerable plaque components showed a good correlation with pathological specimens [7]. A study found that the prevalence of symptomatic plaque is significantly higher in patients with IPH, regardless of how long a neurologic event occurs [4]. Comparing the area of high intensity in the plaque surrounding the carotid artery with histopathologic findings, the sensitivity, specificity, positive predictive value, and negative predictive value of intraplaque hemorrhage on TOF images were 91%, 83%, 72%, and 95%, respectively [13]. The presence of IPH in the carotid atherosclerotic plaque is an independent risk factor for stroke. These findings indicate the promise of IPH as a marker of plaque vulnerability in healthy individuals with subclinical atherosclerosis [33]. For fibrous cap rupture (FCR), the difference between symptomatic and asymptomatic patients is significant in the first 15 days after the neurologic event [4]. Evaluation of the preoperative appearance of the fibrous cap in a study of patients undergoing endarterectomy had high test sensitivity (81%) and specificity (90%) for identifying unstable caps in vivo [25]. The largest-ever histological Oxford Plaque study included detailed, reproducible histological assessments of the nature and timing of the onset of symptoms, reporting a high incidence of fibrous cap rupture, a large lipid-rich necrotic core, dense macrophage infiltration, and various degrees of intraplaque hemorrhage [34,35]. Recent studies have demonstrated the clinical utility of HRMRI in the management of extracranial carotid atherosclerosis. For example, HRMRI can be used to identify high-risk plaques that require more aggressive treatment, such as carotid endarterectomy or stenting [36]. HRMRI can also be used to assess the natural history of plaque progression or to monitor the effectiveness of drug therapy. Subsequent sequential imaging will provide information about the time course of atherosclerosis and the effects of treatment. This could enhance the clinical significance of this novel imaging and improve stroke treatment options. Noninvasive identification of lipid cores may have important applications in lipid-lowering clinical trials. In several clinical trials [7,37,38,39,40,41], noninvasive HRMRI results (lipid core or plaque volume) were used as endpoints to evaluate the effect of statin therapy or to monitor the effect of different doses on plaque volume and composition. HRMRI can improve the understanding of the pathophysiology and diagnosis of carotid atherosclerosis and provide greater benefits for the prevention of recurrent stroke in patients by selecting appropriate surgery, endovascular intervention, and optimal drug therapy (intensive risk factor control and antiplatelet therapy).

## 3. HRMRI of Intracranial Atherosclerotic Disease

Intracranial atherosclerotic stenosis (ICAS) is one of the main causes of ischemic stroke and is closely associated with high incidence and mortality of stroke [42]. Thus, noninvasive detection of morphological features can have significant clinical implications in identifying and interpreting high-risk plaques before clinical events occur [13]. Without detailed knowledge of plaque location, severity, and morphology, the mechanism of stroke cannot be directly determined [7,43]. However, HRMRI can complement this information. HRMRI can identify the major tissue components of carotid atherosclerotic plaques by 3D-TOF MRA localization scan reconstruction and comparative analysis of 3D T1WI, 3D T2WI, CE-T1WI and other sequences and enable automatic identification of plaque components (Figure 2).This direction of research is important, as it enables clinicians to quantitatively measure blood vessels, lumen, plaque, lipid nucleus, fiber composition, and plaque segmentation, and obtain repeated measurements of plaque volume. This helps improve clinical guidance, alert clinicians to the possibility of plaque instability, and intervene decisively to reduce patient morbidity and mortality. More and more clinical studies have shown that HRMRI is of great significance for the identification of imaging biomarkers of recurrent ischemic cerebrovascular events [44,45,46]. In the past few decades, significant progress has been made in clinical studies of intracranial arterial calcification (IAC) due to HRMRI [47]. IAC in the intracranial internal carotid artery (ICA) has been shown to be an independent risk factor for ischemic stroke, accounting for 75% of all strokes [48]. A study found that the hyperintensity of T1WI in the culprit plaque of sICAS was independently associated with recurrence in stroke patients after six months [49]. Previous studies on carotid plaques showed that the hyperintensity of the T1WI often represented IPH or lipid nucleation, which increased the risk of plaque rupture and the occurrence of arterial embolism resulting in stroke recurrence. Plaque enhancement is a reliable imaging biomarker [3,44]. The degree of plaque enhancement may reflect the level of inflammatory activity caused by increased endothelial permeability and new blood vessels [46]. It can be a marker of inflammation, plaque instability, and new blood vessels. Identifying and classifying the levels of lipid core and fibrous components in plaques is the most important measure for predicting the likelihood of worsening vascular fragility and stenosis. Pathogenic intracranial atherosclerotic plaques had more contrast enhancement (CR) and CR ≥ 53 had a 78% sensitivity for detecting culprit plaques and a 90% negative predictive value [50]. HRMRI can provide valuable information on middle cerebral artery (MCA) plaques [6,51] and that MCA atherosclerosis may share an underlying pathophysiology with carotid atherosclerosis [18,27,43]. The ability to observe a high signal on T1-weighted images (HST1) in MCA plaques, associated with ipsilateral stroke, provides significant clinical significance for HRMRI. HRMRI can provide insight into the underlying vascular biological differences between symptomatic and asymptomatic MCA stenoses, which can help stratify stroke risk and modify treatment strategies. For the posterior circulation, rupture FC and IPH of basilar artery plaques are independent risk factors for acute infarction [52]. One study [22] found that some patients with posterior circulation infarction had vertebrobasilar artery atherosclerosis on HRMRI, but digital subtraction angiography (DSA) was normal, suggesting that infarction may be due to penetrating artery disease secondary to vertebrobasilar artery atherosclerosis. Multi-contrast HRMRI techniques can provide simultaneous imaging of the different plaque components, allowing for more accurate characterization of the plaque [53]. In conclusion, HRMRI is a novel imaging technique that enhances our understanding of the mechanisms of intracranial artery stenosis and subsequent infarction. It can help identify high-risk plaques and arterial-to-artery embolic infarcts, thus predicting infarct patterns and providing valuable insights for clinical decision-making [5]. Because endarterectomy specimens are not suitable for ICAS, it is difficult to compare in vivo HRMRI and postoperative histopathological images [54]. To date, only a few studies have been conducted in a limited number of samples, focusing on pathological validation of intracranial HRMRI using postmortem arterial specimens [55]. Intracranial atherosclerosis studies are also susceptible to the effects of stroke stage, MRI technique and plaque characteristic measurement criteria, which may lead to the heterogeneity of results. ICAS is much thinner and more difficult to access and measure than extracranial atherosclerosis, image quality is worse. HRMRI does not sensitively identify intracranial fibrous cap fractures or ruptures. It can be assumed that imaging findings of plaque rupture, even if identified, may have limited positive predictive value for stroke. Considering the high recurrence rate of symptomatic ICAS, more prospective longitudinal studies are needed to explore the best imaging biomarkers to predict early post-stroke deterioration in patients with ICAS [56].

## 4. Other Vascular Wall Diseases

Head and carotid artery stenosis can be caused by a variety of conditions, including atherosclerotic stenosis, dissection, inflammation, vasospasm, and arteriovenous malformations. HRMRI can noninvasively distinguish the underlying pathology of stenosis by identifying plaque components or unique enhancement patterns. Carotid dissection is the most common cause of stroke in young adults [57,58,59]. HRMRI can be used to evaluate acute carotid dissection by providing a detailed description of the structural features of the wall and lumen. HRMRI has been particularly useful in analyzing the partial eccentric vascular lumen and vascular wall of dissected arteries, including occlusion, stenosis, luminal thrombus, crescent wall hematoma, pseudoaneurysm, double lumen, and intimal tear. In addition, HRMRI can help distinguish between dissection and atherosclerosis in cases of carotid artery occlusion [60,61,62,63]. MMD is an idiopathic bilateral internal carotid artery (ICA) terminal stenosis that leads to chronic hypoperfusion and cerebral hemorrhage risk. HRMRI can differentiate the diagnosis of MMD by identifying pathological changes, including intimal fiber cell thickening, excessive proliferation of the vessel wall, active angiogenesis, matrix accumulation, a small number of inflammatory cells in histological studies, irregular fluctuation of the internal elastic layer, medium attenuation, and a decrease in the external diameter of the vessel. HRMRI also provides new insights into the physiopathology of inflammatory arterial disease [64,65]. Imaging analysis of carotid artery stenosis in patients with chronic granulomatous vasculitis has been performed, such as Takayasu arteritis (TA) [66,67] or giant cell arteritis [68,69]. The imaging mode of the artery wall is different from that of atherosclerosis and has a significant inflammatory component. Therefore, it can be used for differential diagnosis in difficult cases. In inflammatory diseases, HRMRI with contrast enhancement can also be used to detect early arterial wall changes, facilitating routine monitoring and assessment of disease activity at a more treatable stage before morphological changes can be detected by other imaging studies. HRMRI can also distinguish between atherosclerosis and basilar artery hypoplasia by identifying plaque components and between atherosclerosis and other rare causes of arterial stenosis or occlusion [32,70,71,72,73,74]. When multiple intracranial aneurysms occur in patients with acute subarachnoid hemorrhage, the incidence of aneurysm may not be easy to identify. However, HRMRI can be used to recognize ruptured aneurysms via contrast enhancement of the vessel wall with high sensitivity. This finding possibly reflects more intense inflammatory changes after aneurysm rupture and highly aneurysmal subarachnoid hemorrhage associated with the risk of vasospasm. Vascular wall imaging also plays an important role in predicting vasospasm caused by noninflammatory processes (reversible cerebral vasoconstriction syndrome, RCVS) [8,9]. In a study evaluating lesions in atherosclerosis, vasculitis, and RCVS, the addition of vessel wall imaging to lumen imaging increased the diagnostic accuracy from 43.5% to 96.3% compared to lumen imaging alone. In addition, 91% of atherosclerotic lesions were found to have an eccentric enhancement pattern, as opposed to concentric enhancement in vasculitis or RCVS [75]. In summary, HRMRI is a valuable tool for noninvasive diagnosis and monitoring of various arterial diseases, providing detailed information on the structural and pathological changes of the vessel wall and lumen. Its ability to differentiate between different pathologies and distinguish between early and advanced disease stages can greatly aid in the management and treatment of these conditions.

## 5. Comparison with Other Imaging Tools

The diagnostic evaluation of head and carotid artery stenosis includes transcranial color-coded duplex ultrasonography(TCCD) and transcranial doppler (TCD), CT angiography (CTA), MRA, and digital subtraction angiography (DSA) [76]. These methods are essential for the accurate diagnosis of cerebrovascular diseases. While DSA remains the gold standard for assessing carotid artery stenosis [77,78], its invasive nature poses a risk to patients. Conventional imaging techniques focus on the vascular lumen, patency, and stenosis, with conventional MR indirectly assessing the degree of lumen stenosis by measuring blood flow velocity. However, stenosis can result from a range of pathologies, including atherosclerosis, inflammation, and vasospasm, and different pathologies can lead to similar stenosis patterns with varying therapeutic implications. HRMRI provides a noninvasive means of directly visualizing the vascular wall in vitro, allowing for the evaluation of luminal stenosis and the pathological features of the vascular wall. HRMRI can also identify vascular remodeling, which may be clinically important but is not recognized by conventional imaging. TCD is a rapid, non-invasive diagnostic technique to determine the degree of stenosis based on blood flow velocity in the cerebral vessels [79]. In the Intracranial Atherosclerosis trial (SONIA) [80], TCD and MRA had high negative predictive values (86 to 91%) but low positive predictive values (36 to 59%) in stroke outcomes and neuroimaging. These data suggest that TCD and MRA can be used for screening to exclude ICAS but not for reliably diagnosing it. CTA is more accurate than MRA in the diagnosis of ICAS, with higher sensitivity and specificity for detecting ≥50% of ICAS. Additionally, HRMRI has been shown to accurately identify lipid-rich necrosis and recent IPH with greater overall accuracy than previously reported, as well as reliably detect fresh hemorrhage [7,8,75,76]. However, there are many limitations in the use of HRMRI. The reliability of HRMRI in detecting high-risk plaque features in ICAS is uncertain and influenced by operator, image sequence, and contrast agent development. At present, the results may be different in different situations and cannot be the only diagnostic method. Distinguishing between symptomatic and asymptomatic plaques, assessing plaque vulnerability, and distinguishing true and pseudo-stenosis all require consideration to improve HRMRI accuracy and repeatability through standardized reporting systems and consensus standards. Currently, HRMRI has become a useful tool for clinical research. Compared with traditional brain angiography, HRMRI provides more valuable pathophysiological information, which is helpful for the differential diagnosis of head and carotid atherosclerosis, dissection, and vasculitis. However, plaque components found on HRMRI may have limited positive predictive value for future stroke [81,82]. The prevalence of plaque rupture in asymptomatic patients may be higher than expected. If vascular dissection and other factors are excluded, partial vascular wall lesions are likely to represent atherosclerosis, and the morphological characteristics of the vascular wall are still useful and reliable diagnostic clues. However, for concentric vascular wall lesions, both atherosclerotic and nonatherosclerotic, the underlying pathophysiology is complex. In conclusion, HRMRI complements traditional luminal and perfusion studies in the assessment of cerebrovascular disease, potentially leading to early diagnosis and treatment, distinguishing disease processes, and facilitating longitudinal assessment. The broad range of clinical applications of HRMRI can provide important clinical insights and improve diagnostic accuracy in high-risk patients, ultimately informing management decisions.

## 6. Future Development of HRMRI

HRMRI is a promising tool for identifying histological features of vulnerable plaques with high reliability and sensitivity. Its ability to detect carotid and cranial artery lesions caused by various factors and to provide valuable pathophysiological information makes it a useful tool for differential diagnosis of head and carotid atherosclerosis, dissection, and vasculitis. The development of HRMRI holds promise for improving stroke treatment options through enhanced risk stratification and personalized treatment [20,21,83,84,85]. In addition to assessing the degree of stenosis and collateral circulation, HRMRI can noninvasively identify other predictors of recurrent symptoms, such as vulnerable plaques, which may be missed conventional vascular imaging [6]. HRMRI can directly assess plaque pattern, including identing the degree of plaque enhancement, the status of tube wall remodeling and the correlation between plaque and recent vascular events, provide a complementary method for early diagnosis of intracranial and extracranial arterial stenosis and targeted treatment of high-risk atherosclerotic lesions. New imaging techniques can provide atherosclerotic information and predict infarct patterns. Appropriate treatment plans can be adopted by judging the mechanism of stroke, such as plaque rupture with local occlusion or interatrial embolism, overgrowth of perforating artery orifice plaque, and stenotic artery hypoperfusion. It can also be used to identify ruptured aneurysms and facilitate surgical decision-making. HRMRI has developed from simple sequences to the assessment of the degree of vascular stenosis to multi-contrast sequences of black blood and bright blood, including 3D-TOF-MRA, T1WI, T2WI, PDWI, CE-T1WI, black blood magnetic resonance imaging (BBMRI) [86,87], and magnetic sensitive weighted imaging. The 1.5 T, 3.0 T to 7.0 T MRA [54], and multi-planar whole-cerebral vascular imaging through 2D imaging technology to 3D imaging technology can improve the accuracy of diagnosis. However, due to the deep location of intracranial arteries, small diameter of tubes and the limitation of magnetic resonance resolution, it is still difficult for HRMRI to accurately describe the components of intracranial arterial plaques. This can be carried out through a variety of different T1-weighted sequences. This is essential to enhance the visibility of any plaque features or enhancements [88]. Multiple tissue weights were performed to assess specific T1 and T2 features to distinguish between plaque components. In addition, most of the current small studies are affected by human, mechanical and other factors, and there is no standard HRMRA imaging protocol for intracranial atherosclerotic diseases. There is also no gold standard model for comparing MR Sequences. A plaque model consisting of narrow blood vessel walls and plaque components has been successfully constructed for the standardization of multi-center HRMRI [89]. However, there are limitations, including the lack of all ICAS plaque components in the prosthesis, particularly intraplaque hemorrhage and calcification seen in atherosclerotic disease. So, large prospective studies are needed to confirm the criteria shown on HRMRI, and further research is necessary to determine its predictive power for treatment and the positive predictive value of its findings for future stroke [6,90]. For the HRMRI process, there will be blood flow artifacts or intimal calcification shade. Multispectral magnetic resonance and multiple sequence plans could become the standard of plaque characterization in vivo. The accuracy and reproducibility of interpreted images will be improved by developing more objective image analysis tools and new tools for the quantitative measurement of cap thickness and volume [7,25]. Perhaps the use of machine learning and artificial intelligence to improve the accuracy of image interpretation and diagnosis will be a big trend in the future. In addition, innovative imaging agents for HRMRI are being developed. If intelligent contrast agents that are more conducive to the assessment of vascular lesions are produced and verified in larger studies, they will provide a broader field for the development of HRMRI in the future [57]. Therefore, it is important to note that HRMRI should not be considered the only diagnostic tool, and a diagnosis should be made based on multiple factors, including patient history, laboratory work, other imaging techniques, and even genetic tests.

## Figures and Tables

**Figure 1 brainsci-13-00677-f001:**
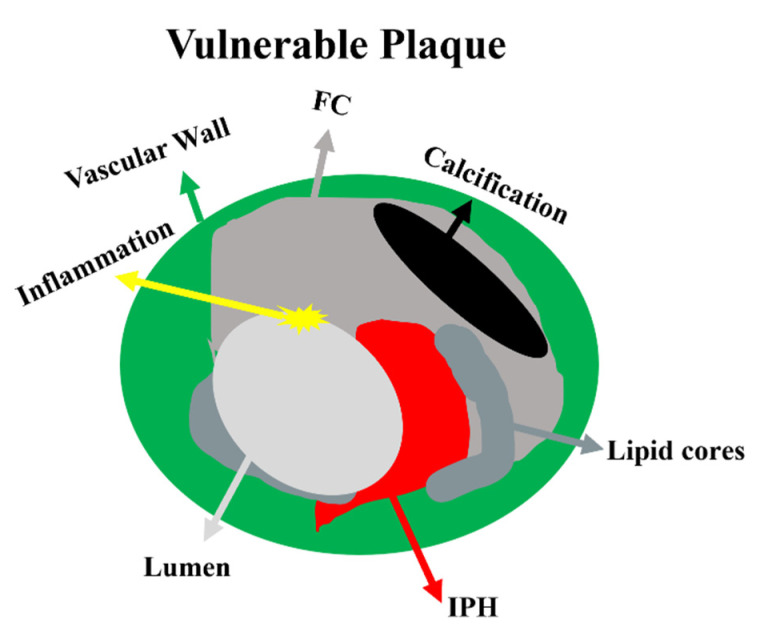
Schematic representation of vulnerable plaque components. Atherosclerotic stenosis is always an eccentric vascular lumen. Vulnerable plaque contains a large lipid-rich necrotic core, IPH, a thin or ruptured FC, or an inflammatory component. However, stable plaques are predominantly fibrous, with small or no lipid cores, few signs of inflammation, and thick continuous fibrous caps. Abbreviations: FC—fibrous cap; IPH—intraplaque hemorrhage.

**Figure 2 brainsci-13-00677-f002:**
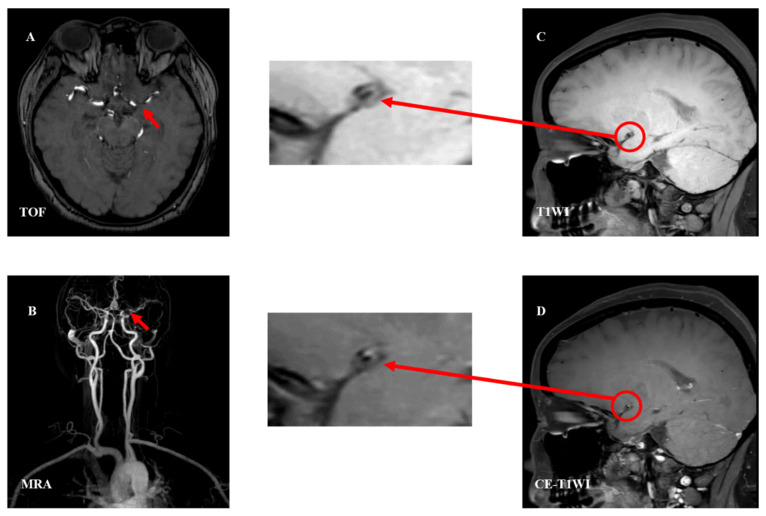
Analysis of examples of intracranial vascular stenosis. The plaque studied were performed on 3.0-T MR units. TOF imaging shows lumen anomalies and localizing stenosis of M1 segment of MCA (panel (**A**), red arrow). Contrast-enhanced MRA development can identify the stenosis (panel (**B**), red arrow). TIWI (panel (**C**), red circle) and CE-T1WI (panel (**D**), red circle) were significantly enhanced and partially reduced, showing characteristics of eccentricity, considering atherosclerotic stenosis with FC and IPH. Abbreviations: TOF—time-of-flight; MRA—magnetic resonance angiography; T1WI—T1-weighted imaging; CE-T1WI—contrast enhanced T1-weighted imaging; FC—fibrous cap; IPH—intraplaque hemorrhage.

**Table 1 brainsci-13-00677-t001:** Plaque components signal.

Plaque Components	TOF	T1WI	T2WI	PDWI	CE-T1WI
Lipid cores	isointensity	isointensity/ hyperintensity	hypointensity	isointensity/ hyperintensity	-
Fibrous caps	isointensity	isointensity/ hyperintensity	isointensity/ hyperintensity	isointensity/ hyperintensity	+
Intraplaque hemorrhage					
Fresh (<1 w)	hyperintensity	hyperintensity	isointensity/ hypointensity	isointensity/ hypointensity	-
Recent (1–6 w)	hyperintensity	hyperintensity	hyperintensity	hyperintensity	-
Chronic (>6 w)	hypointensity	hypointensity	hypointensity	hypointensity	-
Calcification	Heavy hypointensity	Heavy hypointensity	Heavy hypointensity	Heavy hypointensity	-

Abbreviations: TOF—time-of-flight; T1WI—T1-weighted imaging; T2WI—T2-weighted imaging; PDWI—proton density-weighted imaging; CE-T1WI—contrast enhanced T1-weighted imaging.

## Data Availability

The datasets generated during and/or analyzed during the current study are available from the corresponding authors upon reasonable request.

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
