# Peer review of "The Role of High-Resolution Magnetic Resonance Imaging in Cerebrovascular Disease: A Narrative Review"

_brainsci, 2023, doi:10.3390/brainsci13040677_

Round 1

Reviewer 1 Report

Figure 1 is not clear, please add the legends for the table, associated with figure 1.

Authors should explain what is the HRMRI, what are the components, or what kind of MRI sequences are included in the HRMRI, please tabulate.

Authors should provide the table of relevant HRMRI studies along with standard parameters include in the previous studies, and main findings, and on that basis discussion should be modified.

Authors should include some findings correlating HRMRI with histological features, sensitivity, and specificity.

Figure 2 any specific explanation why authors have C and D subfigures first, they can label them as A and B? And do authors have better figures C and D?

Reviewer 2 Report

Dear authors,

thank you for important and nicely written review.

My minor suggestions for improvement:

1) lines 17, 45: stenosis occlussion > stenosis or occlusion

2) Figure 2: could you please provide the explanations (annotations) for images A-B-C-D

3) line 192: I suggest to add methods for neck arteries assessment: "color-coded duplex ultrasonography and transcranial doppler (TCD)..."

4) I suggest uniform format for refferences: capitalize each word (as for refs. 8, 10, 15, etc) vs. capitalize only the 1st word of the sentence (as for refs. 1, 2, 3, etc.).

This manuscript is narrative review, however, could you please provide short description of your strategy for the search of the literature (keywords, databases, etc.)?

Thank you.

Reviewer 3 Report

The authors present a review of high resolution MRI in cerebrovascular disease. Overall this is an important topic that is worthwhile to review. The authors have done a good job beginning to put the review on paper however, it contains considerable issues. Overall the manuscript is hard to follow, disjointed, and theories about vascular disease are split between sections. We suggest a complete restructuring of the review as follows"

I. Introduction

II. Background/science of HRMI

III. Extracranial carotid disease

A. HRMI Plaque/wall findings (per literature review)

B. Clinical implications for findings (per literature review)

C. Clinical recommendations based on findings (per literature review)

IV. Intracranial atherosclerosis 

A. HRMI Plaque/wall findings (per literature review)

B. Clinical implications for findings (per literature review)

C. Clinical recommendations based on findings (per literature review)

V: aneurysms; VI: arteritis/vasculitis/MMD; VII: vasospasm; VIII: other

Round 2

Reviewer 1 Report

No further comments

Author Response

Thank you for your comments concerning our manuscript. Those comments are valuable and very helpful.

Reviewer 3 Report

The authors have done an exceptional job in restructuring and expanding this review. It is much easier to follow and the additional material helps to strengthen the findings. They are to be congratulated on an interesting review. 

Author Response

Thanks for your cafeful, valuable and helpful comments about our manuscript. Your positive feedback improves the quality of the manuscript.